# MAS-SAT: Synergizing ML-Assisted and Standalone Solvers for SAT Solving

## Abstract

Machine learning (ML) has emerged as a promising approach to accelerate the solving of Boolean satisfiability (SAT), with prior research exploiting graph neural networks (GNNs) either as standalone ML solvers or to assist existing SAT solvers. Despite their contributions, the two paradigms mainly suffer from poor scalability and limited compute budgets, respectively. To address these challenges, we propose **M**achine learning **A**ssisting and **S**olving **SAT** (MAS-SAT), a novel framework that synergizes the strengths of both paradigms while mitigating their weaknesses. In MAS-SAT, standalone ML solvers demonstrate improved scalability when solving sub-problems generated by ML-assisted solvers, while ML-assisted solvers achieve both reduced computation by leveraging the parallel search ability of standalone ML solvers, and improved performance from joint training with standalone ML solvers. In addition, we develop an efficient asynchronous deployment strategy with influenced heuristics to further hide the inference time while exploiting the neural heuristics. Extensive experiments across diverse datasets and GNN architectures demonstrate that MAS-SAT consistently outperforms both paradigms. When deployed, MAS-SAT achieves up to $2.4\times$ median speedup on hard instances and solves 3 more instances on SAT Competition 2023 compared to the base state-of-the-art solver, kissat.

## 1 Introduction

Boolean satisfiability (SAT) is a fundamental problem in both theoretical and applied computer science. It was the first problem shown to be NP-complete (Cook, 1971), and its state-of-the-art solvers, conflict-driven clause learning (CDCL) solvers (Marques-Silva et al., 2021), are inherently sequential search algorithms with exponential worst-case complexity. Machine learning (ML) research for SAT has diverged into two paradigms, *standalone ML solvers* that use models to *directly solve* SAT, and *ML-assisted solvers* that integrate models as neural heuristics to *assist* existing solver.

Standalone ML solvers predict SAT solutions in parallel rather than performing sequential searches (Selsam et al., 2018; Amizadeh et al., 2018; Ozolins et al., 2022; Shi et al., 2023). Although successful on small satisfiable instances (with up to around 100 variables), they are incomplete as they cannot guarantee solutions for satisfiable instances or proofs for unsatisfiable instances. Moreover, their success rates drop sharply when instance size scales up (Selsam et al., 2018; Ozolins et al., 2022), limiting their practical application.

ML-assisted solvers integrate neural heuristics into existing SAT solvers (Selsam & Bjørner, 2019; Kurin et al., 2020; Zhang et al., 2021; Wang et al., 2023), thus inheriting their completeness and scalability. However, they remain restricted to sequential searches and incur significant inference overhead, which can be orders of magnitude higher than traditional heuristics. As a result, prior works have allocated only limited budgets to neural heuristics, likely under-exploiting their potential.

In this work, we propose **M**achine learning **A**ssisting and **S**olving **SAT** (MAS-SAT), a novel framework that unifies the two paradigms to demonstrate a synergy. Specifically, we employ graph neural networks (GNNs) as neural heuristics in CDCL solvers while enabling them to operate as standalone solvers. This yields complementary benefits: completeness from CDCL solvers, high-performing neural heuristics from ML-assisted solvers, and parallel solving from standalone ML solvers. More importantly, the two paradigms reinforce each other while mitigating their respective weakness. A

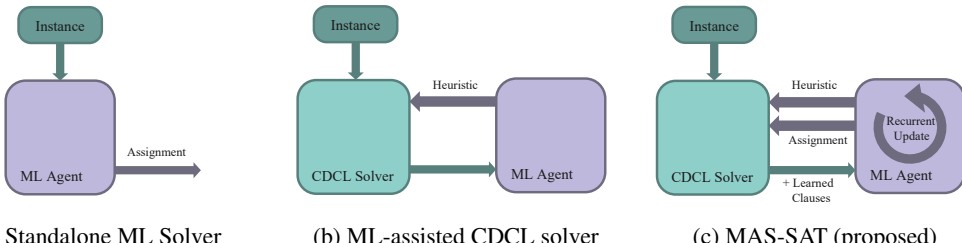

(a) Standalone ML Solver     (b) ML-assisted CDCL solver     (c) MAS-SAT (proposed)

Figure 1: Comparison of paradigms of using machine learning for SAT solving.

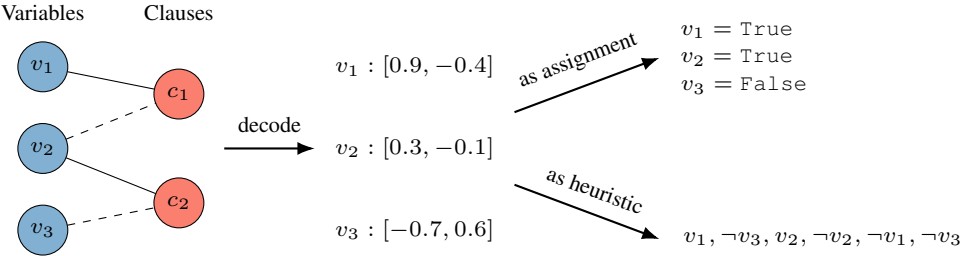

Figure 2: Left: VCG for $\varphi = (v_1 \vee \neg v_2) \wedge (v_2 \vee \neg v_3)$, where solid edges and dashed edges represent positive and negative connections respectively. Middle: an example of logits decoded from variable embeddings. Right: how decoded logits can be used as assignment or heuristic for literal ordering.

visual comparison of the two paradigms and MAS-SAT is presented in Figure 1. To deploy MAS-SAT with actual wall-clock time reduction, we have further proposed an asynchronous deployment strategy with influenced heuristic to hide the inference time of the ML agent while maintaining its guidance to the CDCL solver.

We summarize our contributions as follows:

1. We propose MAS-SAT, a novel framework that synergizes the two paradigms of utilizing machine learning for SAT solving, standalone solvers and ML-assisted solvers. Extensive experiments show that MAS-SAT consistently outperforms both paradigms on a diversity of datasets and GNN architectures.

2. We develop an efficient asynchronous strategy with influenced heuristics for the deployment of MAS-SAT. Deployment experiments show that MAS-SAT can generalize to $10\times$ the instance size as its training set, and is able to accelerate SAT solving in industrial settings, achieving up to $2.4\times$ median relative speedup on hard instances and solves 3 more instances on SAT Competition 2023 (Balyo et al., 2023) compared to the base state-of-the-art CDCL solver, kissat (Fleury & Heisinger, 2020).

MAS-SAT presents an excellent example of how ML techniques can be combined with complex algorithmic approaches. SAT solving has over 50 years of history and kissat (Fleury & Heisinger, 2020) represents the state-of-the-art algorithmic approach with over 38,000 lines of code in C. Yet MAS-SAT is capable of leveraging ML to enhance such a intricate and mature algorithmic system.

## 2 PRELIMINARY

### 2.1 BOOLEAN SATISFIABILITY AND INSTANCE REPRESENTATIONS

SAT determines whether there exists an assignment of truth values (`True` or `False`) to variables of a Boolean formula $\varphi$ that makes it evaluate to `True`. The standard input format to SAT solvers is the conjunctive normal form (CNF) where the formula is a conjunction (AND, $\wedge$) of clauses, each clause is a disjunction (OR, $\vee$) of literals, and each literal is a Boolean variable or its negation (NOT, $\neg$). For example, $\varphi = (v_1 \vee \neg v_2) \wedge (v_2 \vee \neg v_3)$ is a CNF with two clauses $c_1 = v_1 \vee \neg v_2$ and $c_2 = v_2 \vee \neg v_3$, each having two literals. One solution to it is $v_1 = v_2 = $ `True`, $v_3 = $ `False`.

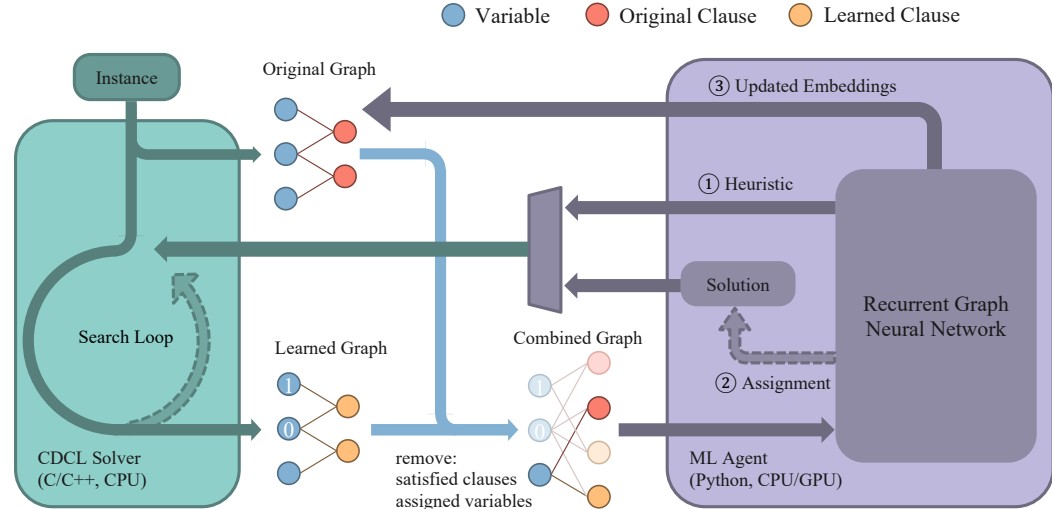

Figure 3: Overview of MAS-SAT with two main components, the CDCL solver and the ML agent.

In learning-based settings, CNFs are commonly represented as variable-clause graphs (VCGs) Guo et al. (2023), which are bipartite graphs consisting of variable nodes and clause nodes. The edges encode the inclusion relationships and their polarities between variables and clauses. An example of VCG is shown in Figure 2. By decoding from the variable embeddings to obtain logits for the positive and negative literals, one can either derive an assignment by comparing the two logits for each variable or a heuristic by sorting the logits. This process is also presented in Figure 2.

## 2.2 CDCL Solvers and Heuristics

The state-of-the-art SAT solvers such as kissat (Fleury & Heisinger, 2020) implement the conflict-driven clause learning (CDCL) algorithm (Marques-Silva et al., 2021). The CDCL algorithm performs a sequential exhaustive search by constantly assigning a truth value to an unassigned variable, propagating its implications to infer the truth values of other variables, until either a satisfying assignment is found or a conflict occurs. In the latter case, the solver learns a new clause by analyzing the conflict to prune the search space and performs backtracking. The introduction of clause learning is crucial to the efficiency and empirical success of CDCL solvers. Pseudo-code of a typical CDCL algorithm is provided in Algorithm A.1 in Appendix C.

Heuristics are widely used in modern CDCL solvers. Two of the most important heuristics are the decision heuristic and the phase heuristic, which respectively determine the variable and the truth value to assign at each step. Researchers have manually designed heuristics such as variable state independent decaying sum (VSIDS) (Eén & Sörensson, 2003) and variable move-to-front (VMTF) (Ryan, 2004), followed by recent efforts that train graph neural networks (GNNs) as neural heuristics (Selsam & Bjørner, 2019; Kurin et al., 2020; Zhang et al., 2021; Wang et al., 2023).

## 3 MAS-SAT

Previous works have successfully built standalone ML solvers (Selsam et al., 2018; Amizadeh et al., 2018; Ozolins et al., 2022; Shi et al., 2023) or ML-assisted SAT solvers (Selsam & Bjørner, 2019; Kurin et al., 2020; Zhang et al., 2021; Wang et al., 2023). Some of them have also leveraged the connection between the two paradigms by transferring standalone ML solvers as neural heuristics to build ML-assisted solvers (Zhang et al., 2021; Wang et al., 2023). However, to our knowledge, none of them has explicitly integrated the two paradigms and allowed the ML agent to perform both tasks. In this paper, the aim is to demonstrate the mutual benefit (synergy) of such integration.

### 3.1 Framework Overview

An overview of our proposed MAS-SAT framework is shown in Figure 3. The CDCL solver runs a search loop that periodically queries the ML agent for decision and phase heuristics. When doing

so, it encodes the current sub-problem as VCG and passes it to the ML agent. The ML agent takes VCG from the CDCL solver as input to a recurrent GNN and produces heuristic $h$ and assignment $a$ to perform as an integration of the two paradigms. We list three joint benefits from such integration, which will be later verified by empirical evaluation in Section 6.

1. **Improved scalability for the standalone ML solver.** Since the standalone ML solver tackles sub-problems produced by the CDCL solver, which are smaller then the original problem, its success rate is greatly improved compared to solving the whole problem.

2. **Improved performance for the ML-assisted solver.** We empirically found that, because of the connection between the two tasks, the GNN backbone for the ML-assisted solver is better trained after integration, leading to neural heuristics with better performance.

3. **Reduced computation for the ML-assisted solver.** When the predicted assignment is verified to be a solution, it can be used to guide the CDCL solver towards solving the instance and later inference can be bypassed to save computation.

## 3.2 State Representation

The substantial difference between the VCG representation adopted by MAS-SAT and previous works (Kurin et al., 2020; Selsam & Bjørner, 2019; Shi et al., 2023; Wang et al., 2023) is that we include *learned clauses* in the graph, reflecting more dynamics in the solving process and encoding richer information in the state representation. Just as clause learning is crucial to CDCL solvers, we argue that the inclusion of learned clauses will help the model understand the previous solving trajectory and hence make more accurate predictions. Specifically, as demonstrated in Figure 3, we will build two VCGs, the original graph $\mathcal{G}_o$ that contains clauses in the original CNF formula and the learned graph $\mathcal{G}_l$ that contains clauses learned by the CDCL solver. The two graphs will be merged into a combined graph $\mathcal{G}_c$ to encode all the knowledge from clause learning.

The current *partial assignment* is another aspect of solver dynamics included in our state representation (Kurin et al., 2020). The rationale is that assigned variables and satisfied clauses do not contribute to the solving process unless backtracked, thus can be safely removed. The result is a smaller sub-graph corresponding to the current sub-problem, and its size will continue to shrink as more variables are assigned. This helps the standalone solver scale better to large instances because it has a higher success rate on smaller graphs.

## 3.3 Model Architecture

The proposed MAS-SAT is compatible with any GNN architecture. In this paper, we examine 3 GNN model architectures in G4SATBench (Li et al., 2023b) to process VCG: Graph Convolutional Network (GCN) (Kipf & Welling, 2016), Gated Graph Neural Network (GGNN) (Li et al., 2015) and Graph Isomorphism Network (GIN) (Xu et al., 2018). We add a global attribute to aggregate global information and significantly reduce the diameter of VCGs (Wang et al., 2023). In addition, we perform recurrent updates to reuse knowledge accumulated in previous iterations. Specifically, the embeddings of the original graph $\mathcal{G}_o$ will be inherited into the combined graph $\mathcal{G}_c$.

Denote $h_v$, $h_c$ and $h_\varphi$ to be the embeddings for variables, clauses and global nodes respectively, the heterogeneous message-passing of the 3 GNN architectures have unified formulations as

$$h_v^{(k+1)} = \text{UPD}_v \left( \underset{c \in \mathcal{N}(v)}{\text{AGG}_v} \left( \left\{ h_c^{(k)} \right\} \right), h_v^{(k)}, h_\varphi^{(k)} \right), \tag{1}$$

$$h_c^{(k+1)} = \text{UPD}_c \left( \underset{v \in \mathcal{N}(c)}{\text{AGG}_c} \left( \left\{ h_v^{(k)} \right\} \right), h_c^{(k)}, h_\varphi^{(k)} \right), \tag{2}$$

$$h_\varphi^{(k+1)} = \text{UPD}_\varphi \left( \underset{v \in \varphi}{\text{AGG}_{\varphi,v}} \left( \left\{ h_v^{(k)} \right\} \right), \underset{c \in \varphi}{\text{AGG}_{\varphi,c}} \left( \left\{ h_c^{(k)} \right\} \right), h_\varphi^{(k)} \right), \tag{3}$$

where $\mathcal{N}(\cdot)$ represents the set of neighbors for a node, and UPD and AGG with different subscripts are the update and aggregation functions. The exact formulation of the 3 GNN architectures is detailed in Appendix B.

After message passing, we can predict assignment, heuristic and value using different MLP heads.

$$\text{Assignment } a = \sigma\left(\text{MLP}_a(h_v)\right), \tag{4}$$

$$\text{Heuristic } h = \text{MLP}_p(h_v), \tag{5}$$

$$\text{Value } V = \text{MLP}_v(h_\varphi). \tag{6}$$

## 4 TRAINING

The loss function of MAS-SAT consists of three terms: 1) the assignment loss $\mathcal{L}_a$ for the standalone ML solver, 2) the policy loss $\mathcal{L}_p$ for the ML-assisted solver and 3) the value loss $\mathcal{L}_v$ of the critic model that aids the training of the ML-assisted solver. The total loss function is formulated as

$$\mathcal{L} = w_a \mathcal{L}_a + w_p \mathcal{L}_p + w_v \mathcal{L}_v, \tag{7}$$

where $w_a$, $w_p$ and $w_v$ are scalar weights of the loss terms.

### 4.1 ASSIGNMENT LOSS

For the assignment loss, we adopt the unsupervised loss proposed by Ozolins et al. (2022), which demonstrates the strongest performance on G4SATBench (Li et al., 2023b). The core idea is to relax Boolean logic to continuous functions to enable differentiable optimization of assignments. Given continuous assignment $a \in (0, 1)^{\#\text{variable}}$, the clause value and assignment loss are formulated as

$$\mathcal{V}_c(a) = 1 - \prod_{i \in c^+}(1 - a_i)\prod_{i \in c^-} a_i, \tag{8}$$

$$\mathcal{L}_a(a) = -\log\left(\prod_{c \in \varphi} \mathcal{V}_c(a)\right) = -\sum_{c \in \varphi} \log \mathcal{V}_c(a), \tag{9}$$

where $c^+$ and $c^-$ are the set of indices of positive and negated literals in clause $c$, respectively.

One thing to note is that sub-problems of a satisfiable instance can be unsatisfiable and have different landscapes of the assignment loss. Therefore, we compute the assignment loss on the original graph instead of the combined graph and empirically find the training to be more stable.

### 4.2 POLICY LOSS AND VALUE LOSS

We perform reinforcement learning to train the ML-assisted solver and adopt the advantage actor-critic (A2C) algorithm (Mnih, 2016) to derive the policy loss $L_p$ and the value loss $L_v$. In the reinforcement learning setting, an episode consists of all the variable decisions and phases that the CDCL solver chooses to solve an instance. The action space at each timestep $t$ consists of the positive and negative literals of all unassigned variables. The ML agent determines what variable and what truth value to assign. We design the reward function to be negatively proportional to increment of propagations, which is the basic operation in CDCL solvers

$$r_t = -\beta(\text{prop}_t - \text{prop}_{t-1}), \tag{10}$$

where $\beta > 0$ makes sure that the number of propagations is minimized and SAT solving is accelerated. The pseudo-code of the reinforcement learning algorithm is presented in Appendix C.

## 5 DEPLOYMENT

### 5.1 ASYNCHRONOUS DEPLOYMENT

Previous works allocate very limited compute budgets to ML models by either running inference at intervals (Selsam & Bjørner, 2019), for a few first decisions (Kurin et al., 2020), or only once before solving starts (Wang et al., 2023). We argue that such synchronous deployment in which CDCL solvers stall and wait for the computation of neural heuristics to finish is highly inefficient.

In light of this, we propose *asynchronous deployment* where the CDCL solver and the ML agent run in concurrent threads. The two components will update each other asynchronously, with only

**Algorithm 1** CDCL Solver of MAS-SAT

**Require:** Formula $\varphi$, CDCL solver $\Pi$, interval $\tau$
  $\mathcal{G}_o = \Pi.\text{init\_and\_parse}(\varphi)$
  $\mathcal{T}_a = \text{start\_agent\_thread}(\mathcal{G}_o)$
  **while not** $\Pi.\text{is\_solved}()$ **do**
    $\mathcal{G}_l = \Pi.\text{get\_learned\_graph}()$
    $\mathcal{T}_a.\text{update\_learned\_graph}(\mathcal{G}_l)$
    $h, a = \mathcal{T}_a.\text{get\_latest\_h\_and\_a}()$
    **if** $\varphi.\text{evaluate}(a)$ **is** TRUE **then**
      $\Pi.\text{use\_solution}(a)$
    **end if**
    $\Pi.\text{influence\_and\_step}(h, \tau)$
  **end while**

**Algorithm 2** GNN Agent of MAS-SAT

**Require:** Original graph $\mathcal{G}_o$, GNN $\mathcal{M}$, solver thread $\mathcal{T}_s$
  **while not** $\mathcal{T}_s.\text{is\_finished}()$ **do**
    $\mathcal{G}_l = \mathcal{T}_s.\text{get\_latest\_learned\_graph}()$
    $\mathcal{G}_c = \text{combine}(\mathcal{G}_o, \mathcal{G}_l)$
    $\mathcal{G}_c.\text{init\_embedding}(\mathcal{G}_o)$
    $h, a = \mathcal{M}(\mathcal{G}_c)$
    $\mathcal{T}_s.\text{update\_h\_and\_a}(h, a)$
    $\mathcal{G}_o.\text{update\_embedding}(\mathcal{G}_c)$
  **end while**

minimal locking overhead introduced to maintain data consistency. Although the heuristic does not reflect the latest state, it still has accumulated knowledge from previous recurrent updates. This strategy brings advantage in wall-clock time by hiding the inference time of the ML agent. Note that during the training time and test time for prediction quality, we still adopt a synchronous setup for deterministic interactions between the CDCL solver and the ML agent.

## 5.2 Influenced Heuristics

Although we have effectively hidden the inference time of the ML agent with asynchronous deployment, the ML agent is still inherently much slower than the CDCL solver. Therefore, their interaction remains relatively sparse and the guidance from the ML agent can still be limited. In light of this, we propose to use the neural heuristic to apply influence on the solver implementations, effectively extending its guidance. In this way, the neural heuristic will not only take over the immediate action, but will also impact future actions as the solver uses the influenced heuristics.

We have designed corresponding influence strategy for the two decision heuristics in kissat (Fleury & Heisinger, 2020). For the queue-based VMTF heuristic (Ryan, 2004), we will reorder the queue according to the neural heuristic. For the priority-value-based VSIDS heuristic (Eén & Sörensson, 2003), we will influence the values by factors ranging from $(1, 2]$ according to their ranks in the neural heuristic,

$$\text{VSIDS}'(x) = (1 + 1/\text{Rank}(x)) \times \text{VSIDS}(x). \tag{11}$$

Combining all the techniques mentioned above, the algorithmic descriptions of the CDCL solver and the GNN agent are described in Algorithms 2 and 1, respectively.

## 6 Evaluation

To validate our proposed MAS-SAT framework, we performed extensive experiments on the 7 datasets and 3 GNN architectures in G4SATBench (Li et al., 2023b).

The 7 datasets come from 3 different domains: 1) SR (Selsam et al., 2018) and 3-SAT are random problems, 2) Community Attachment (CA) (Giráldez-Cru & Levy, 2015) and Popularity-Similarity (PS) (Giráldez-Cru & Levy, 2017) are pseudo-industrial problems and 3) $k$-Clique, $k$-Dominating Set ($k$-Domset) and $k$-Vertex Cover ($k$-Vercov) are combinatorial problems. We generate 80k/1k/1k satisfiable instances at the easy level for each dataset as the training/validation/test set. Description of datasets is provided in Appendix A and implementation details are provided in Appendix D.

### 6.1 Comparison with ML-assisted Solvers

First, we compare the performance of MAS-SAT with ML-assisted solvers that are trained without the assignment loss $\mathcal{L}_a$. The comparison is conducted in a synchronous setting to compare the performance of neural heuristics with deterministic behavior. Similar to Kurin et al. (2020), we use the median of relative reduction rate compared to kissat to measure the performance of neural

Table 1: Comparison between MAS-SAT and ML-assisted solvers. The number of GNN message-passing ($n \downarrow$) and the median relative reduction of propagations with respect to kissat ($r \uparrow$) are presented. The best performance of each column is in bold font.

| Arch. | Paradigm | SR | | 3-SAT | | CA | | PS | | k-Clique | | k-Domset | | k-Vercov | | Average | |
|---|---|---|---|---|---|---|---|---|---|---|---|---|---|---|---|---|---|
| | | $n \downarrow$ | $r \uparrow$ | $n \downarrow$ | $r \uparrow$ | $n \downarrow$ | $r \uparrow$ | $n \downarrow$ | $r \uparrow$ | $n \downarrow$ | $r \uparrow$ | $n \downarrow$ | $r \uparrow$ | $n \downarrow$ | $r \uparrow$ | $n \downarrow$ | $r \uparrow$ |
| GCN | ML-assisted | 8.42 | 1.45 | 11.44 | 1.98 | 35.92 | 3.19 | 13.94 | 1.54 | 26.04 | 1.49 | 24.93 | 9.75 | 23.07 | 12.49 | 20.54 | 4.56 |
| | MAS-SAT | **7.12** | 1.51 | 11.44 | 2.18 | 22.40 | **3.24** | 10.36 | 1.72 | 21.12 | 1.68 | 14.88 | 12.21 | 15.14 | 13.08 | 14.64 | 5.09 |
| | $-assignment$ | 8.49 | 1.52 | 13.43 | 2.19 | 41.56 | 2.91 | 14.57 | 1.72 | 31.25 | 1.68 | 17.63 | 12.66 | 15.81 | 13.18 | 21.82 | 5.12 |
| | $-learned$ | 7.50 | 1.51 | 11.46 | 2.17 | 36.10 | 2.93 | 10.11 | 1.71 | 32.28 | 1.58 | 15.32 | 12.21 | 15.05 | 12.90 | 18.26 | 5.00 |
| GGNN | ML-assisted | 8.51 | 1.46 | 14.35 | 2.04 | 37.70 | 3.00 | 14.36 | 1.50 | 31.88 | 1.62 | 21.20 | 13.14 | 23.14 | 11.14 | 21.59 | 4.84 |
| | MAS-SAT | 7.71 | **1.58** | 10.79 | **2.32** | 25.33 | 3.04 | **8.95** | **1.75** | 19.37 | 1.61 | **9.82** | **13.35** | **10.12** | **13.39** | 13.16 | 5.29 |
| | $-assignment$ | 8.94 | **1.58** | 13.59 | **2.32** | 38.31 | 2.96 | 14.28 | 1.75 | 18.51 | 1.55 | 25.27 | 13.27 | 25.82 | **13.39** | 20.67 | 5.26 |
| | $-learned$ | 7.66 | 1.57 | 10.87 | 2.30 | 37.68 | 2.92 | 9.07 | **1.75** | 17.27 | 1.53 | 9.84 | 13.28 | 10.30 | **13.39** | 14.67 | 5.25 |
| GIN | ML-assisted | 8.28 | 1.38 | 13.85 | 2.00 | 35.23 | 3.21 | 12.37 | 1.61 | 30.22 | **1.95** | 24.23 | 10.87 | 24.06 | 11.46 | 21.18 | 4.64 |
| | MAS-SAT | 7.48 | 1.52 | **10.47** | 2.25 | **21.25** | 3.21 | 9.76 | 1.69 | 30.33 | **1.95** | 12.21 | 12.96 | 11.31 | 13.09 | 14.69 | 5.24 |
| | $-assignment$ | 8.57 | 1.52 | 13.10 | 2.24 | 37.95 | 2.96 | 14.45 | 1.69 | 30.41 | **1.95** | 25.18 | 12.78 | 25.58 | 13.08 | 22.18 | 5.17 |
| | $-learned$ | 7.67 | 1.50 | 10.60 | 2.22 | 34.63 | 2.96 | 9.84 | 1.67 | 30.34 | 1.94 | 12.21 | 12.96 | 11.32 | 13.08 | 16.66 | 5.19 |

Table 2: Comparison between MAS-SAT and standalone ML solver. The number of GNN message-passing ($n \downarrow$) and average success rate of assignment prediction ($p \uparrow$) are presented. The best performance of each column is in bold font.

| Arch. | Paradigm | SR | | 3-SAT | | CA | | PS | | k-Clique | | k-Domset | | k-Vercov | | Average |
|---|---|---|---|---|---|---|---|---|---|---|---|---|---|---|---|---|
| | | $n \downarrow$ | $p \uparrow$ | $n \downarrow$ | $p \uparrow$ | $n \downarrow$ | $p \uparrow$ | $n \downarrow$ | $p \uparrow$ | $n \downarrow$ | $p \uparrow$ | $n \downarrow$ | $p \uparrow$ | $n \downarrow$ | $p \uparrow$ | $p \uparrow$ |
| GCN | Standalone | 8.0 | 0.3% | 12.0 | 6.3% | 23.0 | 43.9% | 11.0 | 20.3% | 22.0 | 49.7% | 15.0 | 30.7% | 16.0 | 42.8% | 27.7% |
| | MAS-SAT | **7.1** | 45.9% | 11.4 | 54.5% | 22.4 | **92.4%** | 10.4 | 72.2% | 21.1 | 51.8% | 14.9 | 55.3% | 15.1 | 85.3% | 65.3% |
| | $-solver$ | 8.0 | 0.2% | 12.0 | 2.4% | 23.0 | 0.0% | 11.0 | 0.0% | 22.0 | 0.0% | 15.0 | 0.0% | 16.0 | 0.0% | 0.4% |
| GGNN | Standalone | 8.0 | 0.6% | 11.0 | 5.6% | 26.0 | 44.0% | 9.0 | 13.6% | 20.0 | 45.9% | 10.0 | 19.8% | 11.0 | 15.7% | 20.7% |
| | MAS-SAT | 7.7 | **46.9%** | 10.8 | **58.8%** | 25.3 | 90.0% | **9.0** | **78.2%** | 19.4 | 34.7% | **9.8** | **94.0%** | 10.1 | **93.7%** | 70.9% |
| | $-solver$ | 8.0 | 0.4% | 11.0 | 5.3% | 26.0 | 0.0% | 9.0 | 7.0% | 20.0 | 0.0% | 10.0 | 23.5% | 11.0 | 11.7% | 6.8% |
| GIN | Standalone | 8.0 | 0.5% | 11.0 | 5.5% | 22.0 | 46.4% | 10.0 | 16.7% | 31.0 | **59.1%** | 13.0 | 42.4% | 12.0 | 40.3% | 30.1% |
| | MAS-SAT | 7.5 | 45.6% | **10.5** | **58.8%** | **21.3** | 91.9% | 9.8 | 75.0% | 30.3 | 57.3% | 12.2 | 85.3% | 11.3 | 92.0% | 72.3% |
| | $-solver$ | 8.0 | 0.3% | 11.0 | 5.2% | 22.0 | 0.0% | 10.0 | 3.9% | 31.0 | 0.0% | 13.0 | 31.9% | 12.0 | 7.7% | 7.0% |

heuristics (the higher the better), which is formulated as

$$r = \text{median}\left\{ \text{prop}_{i,\text{kissat}} / \text{prop}_{i,\text{ML-assisted}} \,\middle|\, i = 1, \ldots, |\mathcal{D}| \right\}. \tag{12}$$

We also report the number of GNN message-passing $n$ to measure the computational cost (the lower the better). From Table 1, it can be observed that MAS-SAT consistently outperforms ML-assisted solvers and achieves lower $n$ and higher $r$ in most cases.

The advantage of MAS-SAT over ML-assisted solvers can possibly be attributed to two reasons: 1) once the standalone solver finds a solution, the ML-assisted solver can save inference cost and use the solution to solve the instance; 2) the ML-assisted solver benefit from better GNN backbone jointly-trained as a standalone solver. To further investigate the two possible reasons, we disable the assignment head of MAS-SAT after joint training as an ablation study. The results are presented as the $-assignment$ rows in Table 1. From the ablation results, disabling assignment prediction significantly increases $n$ from 13.16 to 20.67 on average for GGNN, but only slightly decreases $r$ from 5.29 to 5.26. This evidences that the two reasons both hold. While the integration of standalone ML solver brings reduction in $n$, joint training also benefits the performance of neural heuristics and result in significant increase in $r$.

We also perform an ablation study on the inclusion of learned clauses in VCG. When learned clauses are excluded (presented as the $-learned$ rows in Table 1), $n$ increases from 13.16 to 14.67 and $r$ decreases from 5.29 to 5.25 on average for GGNN. Therefore, we conclude that inclusion of learned clauses is beneficial for the ML-assisted solvers.

## 6.2 COMPARISON WITH STANDALONE SOLVERS

Next, we compare the performance of MAS-SAT with standalone ML solvers that are trained with assignment loss $\mathcal{L}_a$ alone. We round up $n$ of MAS-SAT and use that for the standalone ML solvers to ensure fair comparison by allowing them to produce the same number of assignment predictions. For an instance, if any one of the predicted assignment satisfies the formula, we consider it a success. We report the average success rate ($p$) as well as $n$ in Table 2. From the results, it can be observed that MAS-SAT consistently achieves higher $p$ than standalone ML solvers in most cases.

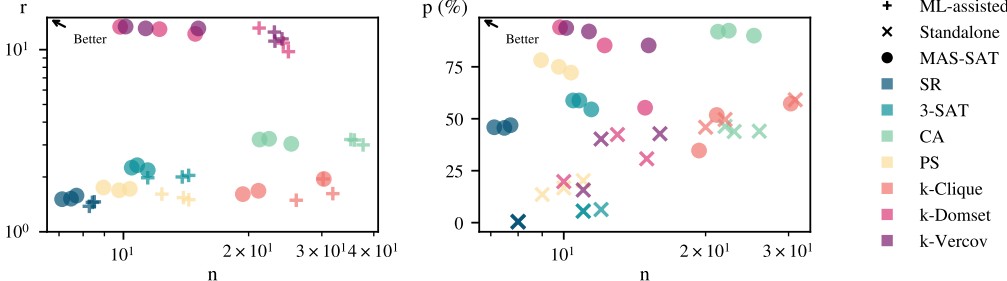

Figure 4: Visual comparison between MAS-SAT and ML-assisted solver (left) and standalone ML solver (right). The horizontal axes represent the number of GNN message-passing ($n \downarrow$). The vertical axes represent the median relative reduction of propagations with respect to kissat ($r \uparrow$) and the success rate of standalone solver ($p \uparrow$), respectively.

Table 3: GGNN deployment results in a synchronous setup. The number of GNN message-passing ($n \downarrow$) and median relative reduction of propagations with respect to kissat ($r \uparrow$)) are presented.

| $\tau$ | Infl. | SR $n \downarrow$ | $r \uparrow$ | 3-SAT $n \downarrow$ | $r \uparrow$ | CA $n \downarrow$ | $r \uparrow$ | PS $n \downarrow$ | $r \uparrow$ | $k$-Clique $n \downarrow$ | $r \uparrow$ | $k$-Domset $n \downarrow$ | $r \uparrow$ | $k$-Vercov $n \downarrow$ | $r \uparrow$ | Average $n \downarrow$ | $r \uparrow$ |
|---|---|---|---|---|---|---|---|---|---|---|---|---|---|---|---|---|---|
| 1 | | 7.71 | 1.58 | 10.79 | 2.32 | 25.33 | 3.04 | 8.95 | 1.75 | 19.37 | 1.61 | 9.82 | 13.35 | 10.12 | 13.39 | 13.16 | 5.29 |
| 2 | ✗ | 13.70 | 1.32 | 23.34 | 1.49 | 118.08 | 1.33 | 16.02 | 1.43 | 65.40 | 1.00 | 99.06 | 2.18 | 129.94 | 1.63 | 66.51 | 1.48 |
| | ✓ | 13.13 | 1.65 | 21.70 | 2.07 | 156.63 | 1.14 | 14.23 | 1.61 | 56.64 | 1.01 | 12.89 | 6.24 | 24.20 | 5.80 | 42.77 | 2.79 |
| 4 | ✗ | 8.52 | 1.16 | 14.44 | 1.26 | 63.76 | 1.15 | 11.16 | 1.29 | 27.02 | 1.00 | 51.78 | 1.61 | 71.11 | 1.40 | 35.40 | 1.27 |
| | ✓ | 8.04 | 1.55 | 13.10 | 1.92 | 90.02 | 0.94 | 9.12 | 1.53 | 26.48 | 1.16 | 23.09 | 5.12 | 15.14 | 4.99 | 26.43 | 2.46 |

This advantage of MAS-SAT over standalone ML solvers can similarly be attributed to two possible reasons: 1) instead of solving the whole instance, the standalone ML solver in MAS-SAT will solve sub-problems produced by the CDCL solver, which are smaller in scale and much easier than the original instances; 2) the standalone ML solver can also benefit from better GNN backbone from joint training. Out of a similar mindset to investigate the two possible reasons, we disable the integration with CDCL solvers to use MAS-SAT as standalone solvers to solve the original instances. The results are presented as the $-solver$ rows in Table 2. Surprisingly, $p$ drops drastically from 72.3% to 7.0% on average for GIN, and is much worse than standalone ML solvers trained alone (30.1% on average). This indicates that in MAS-SAT, the assignment prediction especially learns to cooperate with the CDCL solver and to solve the sub-problems, but not to solve static instances as conventional standalone ML solvers do. That is, only reason 1) holds. Nevertheless, the success rate and scalability of standalone solvers are still improved in the MAS-SAT framework.

From comparisons presented above, both the ML-assisted solvers and the standalone ML solvers benefit from their integration in MAS-SAT, demonstrating a synergy. The results are further visualized in $n$-$r$ and $n$-$p$ planes in Figure 4, where it can be clearly seen that MAS-SAT is moving towards corners with smaller $n$, larger $r$ and higher $p$, which indicate better performance.

### 6.3 EFFECTIVENESS OF INFLUENCED HEURISTIC

To validate the effectiveness of our proposed deployment strategy with influenced heuristic, we have performed ablation studies in a synchronous setup where we gradually increase the interval $\tau$ that the CDCL solver query the ML agent. The results with and without the influence strategy are presented in Table 3. Due to space constraints, only GGNN results are reported and the full results are provided in Appendix E. It can be seen that when $\tau$ is increased to 2 and 4, $r$ drastically degrades from 5.29 to 1.48 and 1.27 respectively. However, with the influence strategy, $r$ can be largely recovered to 2.79 and 2.46 respectively, maintaining the guidance of the neural heuristic.

### 6.4 DEPLOYMENT ON PRACTICAL BENCHMARKS

From evaluation above, it can be seen that GGNN performs the best out of the 3 GNN architectures, which aligns with the conclusions drawn by G4SATBench (Li et al., 2023b). Therefore, we train GGNN on all the training dataset combined and perform asynchronous deployment on practical benchmarks, with the aim of reducing the wall-clock time.

Table 4: Asynchronous deployment of GGNN on the hard instances of G4SATBench. Each setup is performed for 10 rounds and the min, median and max $r$ in terms of wall-clock time is reported.

| Interval $\tau$ | SR | | | 3-SAT | | | CA | | | PS | | | $k$-Clique | | | $k$-Domset | | | $k$-Vercov | | |
|---|---|---|---|---|---|---|---|---|---|---|---|---|---|---|---|---|---|---|---|---|---|
| | min | med | max | min | med | max | min | med | max | min | med | max | min | med | max | min | med | max | min | med | max |
| 1000 | 0.95 | 1.01 | 1.03 | 0.81 | 0.91 | 0.95 | 1.08 | 1.10 | 1.15 | 1.58 | 1.71 | 1.78 | 2.14 | 2.20 | 2.27 | 1.84 | 1.93 | 1.96 | 1.79 | 1.82 | 1.89 |
| 10,000 | 1.16 | 1.21 | 1.24 | 0.83 | 0.93 | 0.98 | 1.29 | 1.34 | 1.41 | 1.88 | 1.93 | 2.16 | 1.52 | 2.17 | 2.25 | 1.84 | 1.89 | 1.97 | 2.07 | 2.17 | 2.20 |
| 100,000 | 1.01 | 1.05 | 1.06 | 0.92 | 0.97 | 0.99 | 1.33 | 1.37 | 1.42 | 1.67 | 2.08 | 2.28 | 2.16 | 2.33 | 2.40 | 1.68 | 1.83 | 1.95 | 1.86 | 1.98 | 2.03 |

First, MAS-SAT is deployed on hard-level instances in G4SATBench (Li et al., 2023b), which can have up to $10\times$ variables compared to the training set. From Table 4, our asynchronous deployment strategy achieves consistent improvement on 6 out of 7 datasets (except on 3-SAT, where MAS-SAT is slightly slower) and can achieve up to $2.4\times$ median relative speedup over kissat. The deployment is relatively robust to the interaction interval $\tau$ between the CDCL solver and the GNN agent.

Then, to demonstrate the performance of MAS-SAT on industrial instances, we further deploy it on the 400 instances from SAT Competition 2023 (Balyo et al., 2023) with the same timeout 5,000s as the competition. Using interval $\tau = 10,000$, MAS-SAT is able to solve 3 more instances than kissat (Fleury & Heisinger, 2020) (264 vs 261). More details can be found in Appendix F.

## 7 RELATED WORKS

NeuroSAT (Selsam et al., 2018) first reported the ability of recurrent GNNs to predict satisfiability and act as standalone solvers when trained with supervision. It also proposed to decode the satisfied assignments by clustering the embeddings. QuerySAT (Ozolins et al., 2022) proposed an unsupervised loss to relieve reliance on labelled data, and a query mechanism to boost the performance of standalone solvers. DG-DAGRNN (Amizadeh et al., 2018) and DeepSAT (Li et al., 2023a) explored other graph representations of SAT (circuit-DAG and AIG, respectively). SATFormer (Shi et al., 2023) explored other model architecture (Transformer) and multi-task learning. The major shortcomings of standalone ML solvers are incompleteness and poor scalability to large instances.

NeuroCore (Selsam & Bjørner, 2019) brings the success of standalone ML solvers to ML-assisted solvers by transferring a model trained for unsatisfiable core prediction as decision heuristics. In a similar vein, SATFormer (Shi et al., 2023) transferred a model trained for satisfiability and unsatisfiable core prediction as decision heuristics and NeuroBack (Wang et al., 2023) transferred a model trained for backbone variable phase prediction as phase heuristics. However, these surrogate tasks only indirectly connect to SAT acceleration, which might lead to sub-optimal heuristics. Graph-Q-SAT (Kurin et al., 2020) first adopted reinforcement learning (RL) to learn decision heuristics.

It is challenging to achieve wall-clock time reduction for ML-assisted solvers because neural heuristics can be orders of magnitudes slower compared to well-optimized heuristics in modern SAT solvers. In face of this challenge, previous works limit the computation budget of the ML agent in various ways. NeuroCore (Selsam & Bjørner, 2019) calls neural heuristics at pre-determined intervals to reduce the frequency of inference. Graph-Q-SAT (Kurin et al., 2020) only consults neural heuristics for a first few decisions. NeuroBack (Wang et al., 2023) only performs inference once before solving. Although these strategies reduce the inference cost, the sparse interaction between solvers and ML agents makes it very likely to under-exploit the ability of neural heuristics.

## 8 CONCLUSION

In this work, we propose **M**achine learning **A**ssisting and **S**olving **SAT** (MAS-SAT), a novel framework that achieves synergy of the two current paradigms of using machine learning to accelerate SAT solving, standalone ML solver and ML-assisted solver. We have also proposed an efficient asynchronous deployment strategy with influenced heuristics to hide the inference time of the ML agent and promote its impact. Experiments on a diversity of datasets and architectures demonstrate that MAS-SAT is able to largely enhance the performance of both paradigms. Compared to the base state-of-the-art CDCL solver, kissat (Fleury & Heisinger, 2020), MAS-SAT has achieved up to $2.4\times$ median speedup in wall-clock time on hard SAT instances and solves 3 more instances on the benchmark for SAT Competition 2023.

## ETHICS STATEMENT

This work does not involve human subjects, sensitive personal data or experiments that raise ethical concerns. We use publicly available codebase and datasets, including kissat (Fleury & Heisinger, 2020) and G4SATBench (Li et al., 2023b). Therefore, we do not foresee any ethical risks. All authors have read and adhere to the ICLR Code of Ethics.

## REPRODUCIBILITY STATEMENT

We have taken care to ensure that all experiments and results presented in this paper are reproducible. Detailed descriptions of datasets, architectures, algorithms, hyperparameters and implementation are included in Appendix A, B, C and D. The codebase is provided in the supplementary material and will be released upon acceptance.

## USE OF LARGE LANGUAGE MODELS (LLMS)

In this work, large language models (LLMs) were used solely as assistive tools for secondary tasks, such as proofreading and the implementation of standard algorithms. They were not involved in generating research ideas or designing novel methods. All outputs from LLMs were manually checked for accuracy and correctness.

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

## A    DATASETS

We provide detailed description of the 7 datesets in G4SATBench (Li et al., 2023b) in Table A.1, along with the statistics of our generated test sets in Table A.2.

## B    GNN ARCHITECTURES

We provide detailed formulation of the message-passing algorithms in the 3 GNN architectures we examined in Table A.3.

## C    ALGORITHMS

We provide the pseudo-code of typical CDCL algorithm in Algorithm A.1 and the Advantage Actor-Critic reinforcement learning algorithm in Algorithm A.2.

## D    HYPERPARAMETERS AND IMPLEMENTATION DETAILS

The hyperparameters used in MAS-SAT is reported in Table A.4.

All of our experiments are performed on server machines with AMD EPYC 7V13 64-Core Processors and AMD Instinct MI210 GPUs. The models are built using PyTorch (Paszke et al., 2019) and PyTorch Geometrics (Fey & Lenssen, 2019).

## E    FULL RESULTS ON THE EFFECTIVENESS OF INFLUENCED HEURISTIC

The full results on all 3 GNN architectures that validate the effectiveness of the proposed deployment strategy with influenced heuristic are presented in Table A.5. The same conclusion as drawn in Section 6.3 holds. Applying influence on solver heuristics effectively extends the guidance of neural heuristics when the interval $\tau$ increases and the interaction between the CDCL solver and the ML agent gets sparse.

## F    DETAILED RESULTS ON SAT COMPETITION 2023

We provide the survival plot and detailed statistics of kissat and MAS-SAT on SAT Competition 2023 Balyo et al. (2023) in Figure A.1 and Table A.6, respectively. Compared to kissat Fleury & Heisinger (2020), MAS-SAT is able to solve 3 more satisfiable instances and reduce the PAR-2 score (penalized average runtime in seconds) by 93.84.

Table A.1: Description of datasets in G4SATBench (Li et al., 2023b)

| Dataset | Description | Parameters |
|---|---|---|
| **Random Domain** | | |
| SR | The SR dataset contains satisfiable and unsatisfiable instance pairs that only differs in polarity of a single literal. Given the number of variables $n$, random clauses with length $k = 1 + \mathrm{Bernoulli}(b) + \mathrm{Geometric}(g)$ and random literals are continuously generated until the instance becomes unsatisfiable. By negating the first literal in the last clause, a satisfiable instances is generated to pair the unsatisfiable one. | General: $b = 0.3, g = 0.4$, same as in NeuroSAT (Selsam et al., 2018), Easy: $n \sim \mathrm{Uniform}(10, 40)$, Medium: $n \sim \mathrm{Uniform}(40, 200)$, Hard: $n \sim \mathrm{Uniform}(200, 400)$. |
| 3-SAT | The 3-SAT dataset contains instances at the phase transition (Gent & Walsh, 1994), where the generated numbers of satisfiable and unsatisfiable instances are roughly the same. Given the number of variables $n$, the number of clauses $m$ is calculated and the clauses are generated by uniformly sampling the literals. | General: $m = 4.258n + 58.26n^{-2/3}$, same as Crawford & Auton (1996), Easy: $n \sim \mathrm{Uniform}(10, 40)$, Medium: $n \sim \mathrm{Uniform}(40, 200)$, Hard: $n \sim \mathrm{Uniform}(200, 300)$. |
| **Pseudo-Industrial Domain** | | |
| CA | The CA dataset (Giráldez-Cru & Levy, 2017) contains SAT instances with a power law distribution in the number of variable occurrences (popularity) and good clustering between them (similarity). Given the number of variables $n$, the number of clauses $m$, the average size of clauses $k$, the exponential parameters $\beta, \beta'$ and the temperature parameter $T$ are sampled. Random angles $\theta_i, \theta_j \in [0, 2\pi]$ are assigned to each variable $i$ and clause $j$, then variables in clauses are randomly sampled with probability $P = 1/(1 + (i^\beta j^{\beta'} \theta_{ij}/R)^T)$, where $\theta_{ij} = \pi - |\pi - |\theta_i - \theta_j||$ and $R$ is a constant that ensures the average size of clauses is roughly $k$. | General: $m \sim \mathrm{Uniform}(6n, 8n)$, $k \sim \mathrm{Uniform}(4, 5)$, $\beta \sim \mathrm{Uniform}(0, 1)$, $\beta' = 1$, $T \sim \mathrm{Uniform}(0.75, 1.5)$, Easy:$n \sim \mathrm{Uniform}(10, 40)$, Medium:$n \sim \mathrm{Uniform}(40, 200)$, Hard:$n \sim \mathrm{Uniform}(200, 300)$. |
| **Combinatorial Domain** | | |
| $k$-Clique | The $k$-Clique problem determines whether there exists a all-connected clique with a specific size in the graph. Given the clique size $v$ and the number of cliques $k$, the edge probability $p$ is calculated according to Bollobás & Erdös (1976) and Erdős–Rényi graphs are generated, which is later converted to SAT instances by CNFGen (Lauria et al., 2017). | General: $p = \binom{v}{k}^{-1/\binom{v}{2}}$, Easy:$v \sim \mathrm{Uniform}(5, 15)$, $k \sim \mathrm{Uniform}(3, 4)$, Medium:$v \sim \mathrm{Uniform}(15, 20)$, $k \sim \mathrm{Uniform}(3, 5)$, Hard:$v \sim \mathrm{Uniform}(20, 25)$, $k \sim \mathrm{Uniform}(4, 6)$. |
| $k$-Domset | The $k$-Domset problem determines whether there exists a dominating set with at most $k$ vertices so that all vertices are either in the set or adjacent to a node in the set. Given $k$ and the number of vertices $v$, the edge probability $p$ is calculated according to Wieland & Godbole (2001) and Erdős–Rényi graphs are generated, which is later converted to SAT instances by CNFGen (Lauria et al., 2017). | General: $p = 1 - \left(1 - \binom{v}{k}^{-1/(v-k)}\right)^{1/k}$, Easy:$v \sim \mathrm{Uniform}(5, 15)$, $k \sim \mathrm{Uniform}(3, 4)$, Medium:$v \sim \mathrm{Uniform}(15, 20)$, $k \sim \mathrm{Uniform}(3, 5)$, Hard:$v \sim \mathrm{Uniform}(20, 25)$, $k \sim \mathrm{Uniform}(4, 6)$. |
| $k$-Vercov | The $k$-Vercov problem determines whether there exists a vertex cover with $k$ vertices so that all edges have at least one endpoint in the set. Given $k$ and the number of vertices $v$, the edge probability $p$ is calculated and Erdős–Rényi graphs are generated, which is later converted to SAT instances by CNFGen (Lauria et al., 2017). | General: $p = \binom{v}{k}^{-1/\binom{v}{2}}$, Easy:$v \sim \mathrm{Uniform}(5, 15)$, $k \sim \mathrm{Uniform}(3, 5)$, Medium:$v \sim \mathrm{Uniform}(10, 20)$, $k \sim \mathrm{Uniform}(6, 8)$, Hard:$v \sim \mathrm{Uniform}(15, 25)$, $k \sim \mathrm{Uniform}(9, 10)$. |

Table A.2: Statistics of the generated test instances

| Difficulty | Statistics | SR | 3-SAT | CA | PS | $k$-Clique | $k$-Domset | $k$-Vercov |
|---|---|---|---|---|---|---|---|---|
| Easy | #Variable | 25.19 | 26.10 | 33.33 | 23.39 | 35.39 | 40.96 | 40.753 |
| | #Clause | 148.49 | 117.86 | 334.67 | 159.30 | 600.13 | 354.98 | 380.01 |
| | kissat propagations | 64.27 | 123.16 | 128.24 | 74.51 | 137.50 | 661.85 | 668.24 |
| Medium | #Variable | 120.64 | 122.98 | 122.39 | 109.817 | 70.30 | 89.56 | 97.46 |
| | #Clause | 658.93 | 525.80 | 1710.84 | 746.18 | 2257.39 | 1690.20 | 2134.22 |
| | kissat propagations | 2229.43 | 6.62e4 | 1302.60 | 1.73e4 | 555.67 | 2962.69 | 3203.78 |
| Hard | #Variable | 295.64 | 250.64 | 300.42 | 249.24 | 113.62 | 136.99 | 176.68 |
| | #Clause | 1592.72 | 1068.20 | 4204.03 | 1693.38 | 5580.48 | 4006.43 | 6971.43 |
| | kissat propagations | 6.71e4 | 1.24e6 | 1.99e4 | 3.58e5 | 1579.46 | 8455.47 | 1.26e4 |

Table A.3: Formulation of GNN architectures

**GCN** (Kipf & Welling, 2016)

Variable $\quad h_v^{(k+1)} = \text{Linear}_v \left( \left[ \sum_{c \in v+} \frac{\text{MLP}_c^+ \left( h_c^{(k)} \right)}{\sqrt{d_c^+ d_v^+}}, \sum_{c \in v-} \frac{\text{MLP}_c^- \left( h_c^{(k)} \right)}{\sqrt{d_c^- d_v^-}}, h_v^{(k)}, h_g^{(k)} \right] \right)$

Clause $\quad h_c^{(k+1)} = \text{Linear}_c \left( \left[ \sum_{v \in c+} \frac{\text{MLP}_v^+ \left( h_v^{(k)} \right)}{\sqrt{d_v^+ d_c^+}}, \sum_{v \in c-} \frac{\text{MLP}_v^- \left( h_v^{(k)} \right)}{\sqrt{d_v^- d_c^-}}, h_c^{(k)}, h_g^{(k)} \right] \right)$

Global $\quad h_\varphi^{(k+1)} = \text{Linear}_\varphi \left( \left[ \sum_{v \in \varphi} h_v^{(k)}, \sum_{c \in \varphi} h_c^{(k)}, h_g^{(k)} \right] \right)$

**GGNN** (Li et al., 2015)

Variable $\quad h_v^{(k+1)} = \text{GRU}_v \left( \left[ \sum_{c \in v+} \text{MLP}_c^+ \left( h_c^{(k)} \right), \sum_{c \in v-} \text{MLP}_c^- \left( h_c^{(k)} \right), h_g^{(k)} \right], h_v^{(k)} \right)$

Clause $\quad h_c^{(k+1)} = \text{GRU}_c \left( \left[ \sum_{v \in c+} \text{MLP}_v^+ \left( h_v^{(k)} \right), \sum_{v \in c-} \text{MLP}_v^- \left( h_v^{(k)} \right), h_g^{(k)} \right], h_c^{(k)} \right)$

Global $\quad h_\varphi^{(k+1)} = \text{GRU}_\varphi \left( \left[ \sum_{v \in \varphi} h_v^{(k)}, \sum_{c \in \varphi} h_c^{(k)} \right], h_g^{(k)} \right)$

**GIN** (Xu et al., 2018)

Variable $\quad h_v^{(k+1)} = \text{MLP}_v \left( \left[ \sum_{c \in v+} \text{MLP}_c^+ \left( h_c^{(k)} \right), \sum_{c \in v-} \text{MLP}_c^- \left( h_c^{(k)} \right), h_v^{(k)}, h_g^{(k)} \right] \right)$

Clause $\quad h_c^{(k+1)} = \text{MLP}_c \left( \left[ \sum_{v \in c+} \text{MLP}_v^+ \left( h_v^{(k)} \right), \sum_{v \in c-} \text{MLP}_v^- \left( h_v^{(k)} \right), h_c^{(k)}, h_g^{(k)} \right] \right)$

Global $\quad h_\varphi^{(k+1)} = \text{MLP}_\varphi \left( \left[ \sum_{v \in \varphi} h_v^{(k)}, \sum_{c \in \varphi} h_c^{(k)}, h_g^{(k)} \right] \right)$

---

**Algorithm A.1** Typical CDCL Algorithm

---

**Require:** CNF formula $\varphi$
$\quad \nu \leftarrow \{\}, dl \leftarrow 0$ // initialize assignment, decision level
$\quad$ **while not** AllVariablesAssigned$(\varphi, \nu)$ **do**
$\quad\quad (x, v) \leftarrow$ PickBranchingLiteral$(\varphi, \nu)$
$\quad\quad \nu \leftarrow \nu \cup \{(x, v)\}, dl \leftarrow dl + 1$
$\quad\quad$ **if** UnitPropagation$(\varphi, \nu) ==$ **CONFLICT then**
$\quad\quad\quad \beta \leftarrow$ ConflictAnalysis$(\varphi, \nu)$ // backtrack level
$\quad\quad\quad$ **if** $\beta < 0$ **then**
$\quad\quad\quad\quad$ **return UNSAT**
$\quad\quad\quad$ **else**
$\quad\quad\quad\quad$ LearnClause$(\varphi, \nu, \beta)$
$\quad\quad\quad\quad$ Backtrack$(\varphi, \nu, \beta)$
$\quad\quad\quad\quad dl \leftarrow \beta$
$\quad\quad\quad$ **end if**
$\quad\quad$ **end if**
$\quad$ **end while**
$\quad$ **return** $\nu$

---

---

**Algorithm A.2** Advantage Actor-Critic

---

**Require:** start time $t_{start}$, policy network $\pi(a|s)$, value network $V(s)$, discount factor $\gamma \in (0, 1)$
**Returns:** policy loss $L_p$, value loss $L_v$
    $t \leftarrow t_{start}, L_p \leftarrow 0, L_v \leftarrow 0$
    Get state $s_t$
    **repeat**
        Perform $a_t$ according to policy $\pi(a_t|s_t)$
        Receive reward $r_t$ and new state $s_{t+1}, t \leftarrow t + 1$
    **until** terminal $s_t$ or $t - t_{start} == t_{max}$
    $R = \begin{cases} 0 & \text{for terminal } s_t \\ V(s_t) & \text{for non-terminal } s_t \end{cases}$
    **for** $i \in \{t - 1, \ldots, t_{start}\}$ **do**
        $R \leftarrow r_i + \gamma R$
        $L_p \leftarrow L_p + \log \pi(a_i|s_i)(R - V(s_i))$
        $L_v \leftarrow L_v + (R - V(s_i))^2$
    **end for**
    **return** $L_p/(t - t_{start}), L_v/(t - t_{start})$

---

Table A.4: Hyperparameters used by MAS-SAT

| Name | Value | Note |
|---|---|---|
| **Model Architecture** | | |
| Hidden dimension $d$ | 128 | |
| Number of MLP layers | 2 | For all MLPs, including in message-passing and in heads. |
| $n$ per call | 1 | |
| Activation function | ReLU | |
| Layer normalization | True | Before every MLP and Linear layer. |
| **Reinforcement Learning** | | |
| Discount factor $\gamma$ | 0.99 | |
| Propagation weight $\beta$ | 0.01 | |
| Environment vector factor | 4 | How many episodes are run together. |
| **Optimization** | | |
| Assignment loss weight $w_a$ | 5.0 | |
| Policy loss weight $w_p$ | 1 | |
| Value loss weight $w_v$ | 0.1 | |
| Batch size | 128 | |
| Learning rate | $10^{-4}$ | |
| Learning rate schedule | cosine annealing | |
| Weight decay | $10^{-8}$ | |
| Optimizer | Adam | (Kingma & Ba, 2014) |
| Adam betas | 0.9, 0.999 | |
| Adam eps | $10^{-8}$ | |
| Gradient clipping | 2.0 | $L_2$ norm is used. |
| Gradient reduction coefficient | 0.8 | Used between GNN message-passing to reduce gradient norm. |
| Validation frequency | 1000 | |
| Total optimization step | 20000 | |

Table A.5: Full experimental results on the effectiveness of influenced heuristic in a synchronous setup. The first three columns are the model architecture, the query interval and whether the influence strategy is used, respectively. The number of GNN message-passing ($n \downarrow$) and median relative reduction of propagations with respect to kissat ($r \uparrow$)) are presented.

| Arch. | Int. | Infl. | SR $n\downarrow$ | $r\uparrow$ | 3-SAT $n\downarrow$ | $r\uparrow$ | CA $n\downarrow$ | $r\uparrow$ | PS $n\downarrow$ | $r\uparrow$ | k-Clique $n\downarrow$ | $r\uparrow$ | k-Domset $n\downarrow$ | $r\uparrow$ | k-Vercov $n\downarrow$ | $r\uparrow$ | Average $n\downarrow$ | $r\uparrow$ |
|---|---|---|---|---|---|---|---|---|---|---|---|---|---|---|---|---|---|---|
| GCN | 1 | | 7.12 | 1.51 | 11.44 | 2.18 | 22.40 | 3.24 | 10.36 | 1.72 | 21.12 | 1.68 | 14.88 | 12.21 | 15.14 | 13.08 | 14.64 | 5.09 |
| | 2 | ✗ | 14.49 | 1.31 | 21.67 | 1.58 | 113.44 | 1.23 | 18.80 | 1.47 | 52.52 | 1.00 | 105.43 | 2.25 | 150.73 | 1.62 | 68.15 | 1.49 |
| | | ✓ | 13.59 | 1.42 | 20.46 | 2.15 | 152.98 | 1.01 | 22.32 | 1.42 | 70.77 | 1.22 | 135.43 | 2.55 | 23.19 | 5.86 | 82.68 | 2.23 |
| | 4 | ✗ | 8.71 | 1.18 | 13.46 | 1.35 | 61.16 | 1.12 | 12.36 | 1.25 | 26.58 | 1.00 | 49.72 | 1.76 | 65.66 | 1.51 | 33.95 | 1.31 |
| | | ✓ | 8.43 | 1.41 | 12.15 | 2.00 | 93.67 | 0.81 | 11.44 | 1.32 | 35.79 | 1.14 | 31.82 | 4.45 | 19.51 | 4.55 | 30.40 | 2.24 |
| GGNN | 1 | | 7.71 | 1.58 | 10.79 | 2.32 | 25.33 | 3.04 | 8.95 | 1.75 | 19.37 | 1.61 | 9.82 | 13.35 | 10.12 | 13.39 | 13.16 | 5.29 |
| | 2 | ✗ | 13.70 | 1.32 | 23.34 | 1.49 | 118.08 | 1.33 | 16.02 | 1.43 | 65.40 | 1.00 | 99.06 | 2.18 | 129.94 | 1.63 | 66.51 | 1.48 |
| | | ✓ | 13.13 | 1.65 | 21.70 | 2.07 | 156.63 | 1.14 | 14.23 | 1.61 | 56.64 | 1.01 | 12.89 | 6.24 | 24.20 | 5.80 | 42.77 | 2.79 |
| | 4 | ✗ | 8.52 | 1.16 | 14.44 | 1.26 | 63.76 | 1.15 | 11.16 | 1.29 | 27.02 | 1.00 | 51.78 | 1.61 | 71.11 | 1.40 | 35.40 | 1.27 |
| | | ✓ | 8.04 | 1.55 | 13.10 | 1.92 | 90.02 | 0.94 | 9.12 | 1.53 | 26.48 | 1.16 | 23.09 | 5.12 | 15.14 | 4.99 | 26.43 | 2.46 |
| GIN | 1 | | 7.48 | 1.52 | 10.47 | 2.25 | 21.25 | 3.21 | 9.76 | 1.69 | 30.33 | 1.95 | 12.21 | 12.96 | 11.31 | 13.09 | 14.69 | 5.24 |
| | 2 | ✗ | 14.58 | 1.27 | 23.27 | 1.47 | 105.44 | 1.36 | 21.38 | 1.40 | 53.33 | 1.00 | 99.17 | 2.12 | 140.26 | 1.53 | 65.35 | 1.45 |
| | | ✓ | 15.54 | 1.39 | 22.30 | 2.04 | 124.49 | 1.32 | 22.36 | 1.54 | 65.98 | 1.25 | 24.60 | 6.11 | 21.44 | 5.91 | 42.39 | 2.79 |
| | 4 | ✗ | 8.69 | 1.12 | 14.86 | 1.25 | 57.39 | 1.23 | 12.83 | 1.23 | 28.12 | 1.00 | 51.20 | 1.72 | 68.69 | 1.44 | 34.54 | 1.28 |
| | | ✓ | 8.26 | 1.31 | 12.81 | 1.91 | 72.98 | 1.12 | 12.34 | 1.46 | 34.20 | 1.22 | 25.48 | 5.20 | 13.42 | 4.99 | 25.64 | 2.46 |

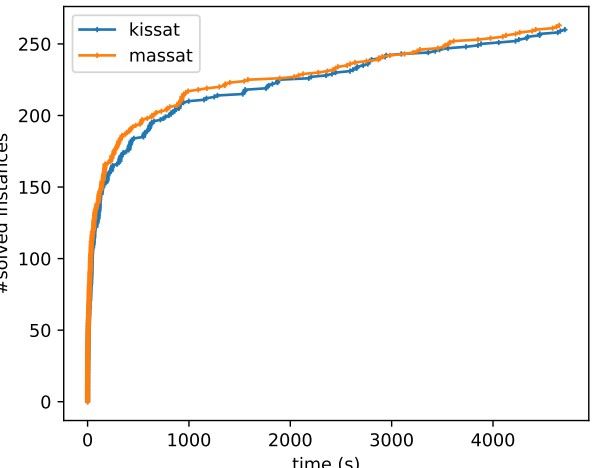

Figure A.1: Survival plot of experiment on SAT Competition 2023.

Table A.6: Detailed statistics on SAT Competition 2023. PAR-2 score is penalized average runtime in seconds where each timeout counts as $2\times$ the time limit, the lower the better.

| | PAR-2 Score↓ | Solved↑ | Solved SAT↑ | Solved UNSAT↑ |
|---|---|---|---|---|
| kissat | 3915.07 | 261 | 119 | 142 |
| MAS-SAT | 3821.23 | 264 | 122 | 142 |

