# OpenReview forum: "MAS-SAT: Synergizing ML-Assisted and Standalone Solvers for SAT Solving"
_ICLR.cc/2026/Conference — Submitted to ICLR 2026_

### Official Review · Reviewer_727G · 2025-10-29

**Soundness:** 3
**Presentation:** 2
**Contribution:** 2
**Rating:** 4
**Confidence:** 3

**Summary:**

The paper proposes MAS-SAT, a novel framework that integrates machine learning (ML) models—specifically graph neural networks (GNNs)—with Conflict-Driven Clause Learning (CDCL) SAT solvers in an asynchronous manner. The key idea is to allow the ML component and the CDCL solver to operate independently but cooperatively: while the CDCL solver proceeds with standard search, the ML model runs in the background to analyze the current formula (or a sub-problem) and provide guidance (e.g., variable assignments or clause predictions) that can be injected back into the solver when ready. This decoupling aims to mitigate the latency bottleneck often seen in tightly coupled ML-augmented solvers.

**Strengths:**

1. The asynchronous integration of ML and CDCL is a creative architectural choice that addresses a real practical limitation—namely, the computational overhead of GNN inference stalling the core SAT solving loop.

2. If the claimed speedups on competition benchmarks hold under scrutiny, this work could influence how future ML-augmented solvers are designed—particularly in high-performance or embedded settings where latency and resource contention matter.

**Weaknesses:**

1. The most critical weakness is the lack of detailed, reproducible experiments on standard benchmarks. The main evaluation relies on a subset of instances from G4SATBench, which are relatively easy (often solvable in seconds). More importantly, the primary metrics reported are proxy measures—number of GNN message-passing steps and median relative reduction in propagations—not actual wall-clock solving time. While propagation reduction is informative, it does not necessarily translate to speedup, especially if GNN inference is expensive.

2. The paper prominently claims performance gains on the SAT Competition 2023 benchmark (“solves 3 more instances”, “2.4× speedup”), but provides little details. Without this, the claim is unverifiable and potentially misleading.

**Questions:**

1. In Figure 1, Type III appears similar to Type II, especially given that BMM-SAT [1] also interleaves ML predictions with CDCL search in a loop. Showing the key differentiating mechanism in MAS-SAT will be helpful.

2. Line 168 mentions extracting a "sub-problem" for the ML model to solve. How is this sub-problem guaranteed to be consistent with the full formula? Could solving it in isolation lead to conflicting assignments when merged back? A brief formalization or example would resolve this concern.

3. Can you provide full experimental details for the SAT Competition 2023 results? Specifically:
- List of instances solved/unsolved by kissat vs. MAS-SAT
- Timeouts, hardware specs, and solver configurations
- Actual wall-clock solving times (not just propagation counts)
- Breakdown of time spent in GNN inference vs. CDCL search

[1] Online Bayesian Moment Matching based SAT Solver Heuristics. ICML 2020.

---

> ### Author Response · Authors · 2025-12-03
>
> Thank you for your thoughtful feedback. We are glad to hear that you find our methodology creative and influential. We hope to address your concerns and questions below.
>
> > [W1] relatively easy benchmarks and proxy metrics.
>
> We would like to clarify that, our experimental evaluation can be divided into two aspects, one on the ML side (Section 6.1 and 6.2) that measures the absolute performance of learned heuristics and assignment predictions, and the other on the deployment side (Section 6.3 and 6.4) that measures the practical speedup using actual wall-clock time on hard competition benchmarks. Relatively easy benchmarks and proxy metrics are used in the first aspect, because they are better suited for synchronous settings and more informative of the quality of learned models.
>
> > [W2/Q3]  detail of the SAT Competition experiments.
>
> The hardware specifications are the same as we listed in Line 624-626 and we use the same 5000s timeout as in SAT Competitions. We have additionally provided the survival plot and detailed statistics of the SAT Competition experiment in Appendix F, where it can be clearly seen that MAS-SAT outperforms kissat by solving 3 more satisfiable instances and reducing the PAR-2 score by 93.84.
>
> > [Q1]  differentiation between Type III and Type II.
>
> To clarify, Type III in Figure 1 is a unification of Type I and Type II, where the ML agent provides both the assignment and heuristic for SAT solving, which essentially integrates the two ML-for-SAT paradigms that were separately used previously. In this categorization, BMM-SAT [1] falls into Type II because the ML agent does not aim to solve the instance directly.
>
> > [Q2] consistency guarantee of subproblems.
>
> The consistency issue is explained in Line 187-191, which we will clarify further here. The subproblems are derived from previous decisions and propagations made by the CDCL solver, which are guaranteed to be correct. As long as we derive a solution of any subproblem, we can get a full solution of the original problem by combining it with previous decisions.
>
> [1] Online Bayesian Moment Matching based SAT Solver Heuristics. ICML 2020.

---

### Official Review · Reviewer_Pip4 · 2025-10-30

**Soundness:** 3
**Presentation:** 3
**Contribution:** 3
**Rating:** 6
**Confidence:** 3

**Summary:**

As we all know, SAT solving is a hard problem and existing solvers rely on heuristics. This one chooses to bolster a specific class of solver - CDCL - to get a hybrid approach that efficiently splits the problem class between what formal, traditional tools are "good at" and what GNNs are "good at", with appropriate passing of heuristics in between the two halves. The result is a hybrid approach that gets us superior performance in quite a few real contest-level benchmarks relative to KISSAT (near-SOTA sat solver).

**Strengths:**

In fields with established non-neural techniques like SAT solving, there are already excellent tools baked with heuristics over the years. In these cases, fresh ML approaches often do not scale because they use those tools as label generators and have to compete in other ways such as runtime, early prediction, etc. The hybrid approaches such as this paper make the most sense there, as it is unlikely that existing users of the field will switch easily.

**Weaknesses:**

There are some drawbacks. In particular, CDCL is not the only class of solver, neither might KISSAT be a one-and-done baseline. SAT contest results will make clear that variants of KISSAT are used in practice, and it is unclear how much this approach adds atop common variant-making techniques like multi armed bandit extensions. Further, a lot of industrial relevance is tied up in extending SAT-level work to SMT-level work such as Z3 which is built around the CDCL core but adds a lot of other tools. How can the paper's approach scale ?

Coming to the empirical results, by far, the most meaningful result is the SAT contest one. Generally speaking, getting only 3 extra instances in a contest for a significant change is somewhat low, and there is no analysis (were >3 instances gained and some dropped ? The runtime is improved - how, uniformly across all the instances ? Or in a peaky way).

There is also no discussion of how this may mean anything for DPLL solvers or for hybrid portfolio methods. Indeed, the paper's right comparison might be to portfolio methods, given that it almost ensembles two things together.

**Questions:**

Can you comment on the weaknesses ? In particular, instead of comparing to kissat, should the comparison not be to the improvement atop kissat which takes effort comparable to the paper, such as a simple 2/3-portfolio of solvers ? How will your approach scale to DPLL solvers / SMT ?

---

> ### Author Response · Authors · 2025-12-03
>
> Thank you for your thoughtful feedback. We are glad that you find our proposed hybrid approach makes sense and easy to switch to. We hope to address your concerns and questions below.
>
> > [W1/Q1] comparison with other variant-making techniques and hybrid portfolio methods.
>
> We have performed additional comparison with top solvers from recent SAT competitions. The solvers are downloaded from https://satcompetition.github.io/. Other setups are the same as the interval=10,000 group in Table 4. Average runtime of three trials in seconds are presented. It can be observed that MAS-SAT still outperforms solvers that adopt MAB as a variant-making technique (kissat_mab and Kissat_MAB-HyWalk).  This is because MAB does not carry cross-instance knowledge and relies on early interactions with the solver to estimate the value of candidate heuristics. In contrast, MAS-SAT can leverage learned cross-instance knowledge and the global graph structure to make good decisions from the beginning.
>
> | Solver              | Note              | SR    | 3-SAT  | CA     | PS     | k-Clique | k-Domset | k-Vercov | Total   |
> |---------------------|-------------------|-------|--------|--------|--------|----------|----------|----------|---------|
> | MAS-SAT             |                   | 19.91 | 544.45 | 19.31  | 125.04 | 11.21    | 11.07    | 10.42    | 741.41  |
> | kissat-3.1.0        | Original baseline | 22.49 | 497.65 | 36.10  | 240.77 | 21.01    | 28.00    | 19.62    | 865.64  |
> | kissat-sc2024   | 2024 winner       | 29.75 | 492.81 | 39.29  | 230.46 | 16.76    | 18.78    | 18.57    | 846.42  |
> | SBVA-Cadical    | 2023 winner       | 64.75 | 1250.92| 11.68  | 263.26 | 4.09     | 4.30     | 12.19    | 1611.19 |
> | Kissat_MAB-HyWalk | 2022 winner     | 20.64 | 494.37 | 42.19  | 272.76 | 18.06    | 17.09    | 16.32    | 881.43  |
> | kissat_mab      | 2021 winner       | 20.58 | 489.63 | 48.91  | 286.78 | 19.09    | 18.30    | 16.19    | 899.48  |
>
> Compared to portfolio approaches that leverage multiple solver instances on multiple CPUs, our approach is inherently different, as we assist one CDCL solver with asynchronous ML inference on GPUs.
>
> > [W2/Q2] scalability to DPLL and SMT.
>
> We would like to clarify that our framework can be integrated with any CDCL-like solvers with branching heuristics. Therefore, it is applicable to DPLL solvers which also rely on branching heuristics, as well as SMT solvers that spawns CDCL solvers internally. The only minor part not applicable to DPLL solvers is the inclusion of learned clauses since DPLL solvers do not implement clause learning.
>
> > [W3] analysis on SAT Competition experiments.
>
> We have additionally provided the survival plot and detailed statistics of the SAT Competition experiment in Appendix F, where it can be clearly seen that MAS-SAT outperforms kissat very uniformly by solving 3 more satisfiable instances and reducing the PAR-2 score by 93.84.

---

### Official Review · Reviewer_gowR · 2025-11-01

**Soundness:** 2
**Presentation:** 1
**Contribution:** 2
**Rating:** 2
**Confidence:** 4

**Summary:**

The paper introduces MAS-SAT, a method that combines ML–assisted and standalone SAT solvers. It uses a graph neural network that can both guide a traditional CDCL solver and directly predict satisfying assignments. The paper describes multiple optimizations to the standalone solver and the ML-assisted solvers by leverage the information shared between the two solvers. Experiments on several benchmark datasets show that MAS-SAT results in overall performance gain on existing benchmarks from the GNN SAT literature and the SAT competition.

**Strengths:**

- The idea to integrate a stand-alone GNN solver and a GNN-assisted SAT solver is novel and interesting.
- The paper describes several ways in which information can be shared between the SAT solver and the GNN to improve the execution of the SAT solver and the efficiency of the GNN. These techniques are novel to me, and empirically, they result in increased performance compared to previous GNN-based solutions.

**Weaknesses:**

- The paper does not report the absolute runtime of the solvers on the evaluated benchmarks, making it difficult to evaluate the significance of the performance gain. If the absolute runtime is small, then even the largest median speed up of 2.4x is not very practically interesting. For SAT competition results, a gain of 3 additional solved benchmarks could simply be noise, especially if there were no significant overall runtime improvement. The paper should include a cactus plot or a scatter plot that compares the actual runtime of the proposed method with that of the baseline method.
- The paper frames the previous work's design choice of "synchronous" execution of the GNN as a limitation of the prior work. However, executing the GNN in the background simply means one devotes more computational resources in parallel, which is relatively straightforward to do and also makes the comparison with Kissat a little more unfair.

**Questions:**

- Could you report the absolute runtime?

---

> ### Author Response · Authors · 2025-12-03
>
> Thank you for your thoughtful feedback. We are glad that you find our integration of two paradigms and proposed techniques novel and practically effective. We hope to address your concerns and questions below.
>
> > [W1/Q1] the absolute runtime and cactus/scatter plot.
>
> We have additionally provided the survival plot and detailed statistics of the SAT Competition experiment in Appendix F, where it can be clearly seen that MAS-SAT outperforms kissat by solving 3 more satisfiable instances and reducing the PAR-2 score by 93.84.
>
> > [W2] devoting more computational resources is an unfair comparison.
>
> We would like to reiterate that, since the CDCL algorithm is inherently sequential, the challenge is less about getting more compute resources, but more about how to better exploit compute resources to accelerate it, especially using GPUs. In our case where GNN is executed in the background, new challenges of sparse and delayed interaction arise. These challenges are non-trivial and we proposed influenced heuristics as a solution in Section 5.2.

---

### Official Review · Reviewer_k2hd · 2025-11-06

**Soundness:** 2
**Presentation:** 2
**Contribution:** 2
**Rating:** 2
**Confidence:** 5

**Summary:**

This paper proposes, MAS-SAT, which is a combination of 1) machine learning assisted sat solving and 2) machine learning as a standalone solver. MAS-SAT is agnostic to underlying machine learning models as long as they are graph neural networks (GNNs). MAS-SAT augments the state representation by incorporating learned clauses by the CDCL heuristic, a standard heuristic used in modern SAT solvers. MAS-SAT runs two solvers (i.e., CDCL solver and ML agent) asynchronously. The experimental evaluations on G4SATBench indicate that MAS-SAT outperforms each individual solver according to the proposed metrics.

**Strengths:**

- this paper explores a hybrid combination of two different styles of ML assisted SAT solvers,  which has not been studied by previous works
- the asynchronous design of two solvers can better leverage compute resources

**Weaknesses:**

- the contribution of this work and reported improvements are due to engineering effort or system design, which is less about machine learning innovations
- slight improvement is somewhat expected, since more compute resources are used in the asynchronous setting
- the idea of augmenting graphs with newly learnt clauses has been studied in the prior work (e.g., G4SATBench), thus not a new contribution
- there are no experimental comparisons with other GNN-based SAT solving
- it is worth noting that portfolio approaches (Balyo et al 2015) are known to be able to outperform individual SAT solvers, and for this very reason, they are forbidden to participate later SAT competitions.

Tomás Balyo, Peter Sanders, Carsten Sinz: HordeSat: A Massively Parallel Portfolio SAT Solver. SAT 2015: 156-172

**Questions:**

What does $n$ the number of GNN message passing mean? Is it the number of interactions between CDCL solver and GNN assisted solver? Similarly, what does $r$ (i.e., relative reduction of propagations with respect to kissat) mean?  Why do they better reflect the performance of SAT solving compared to an obvious metric like running time?

---

> ### Author Response · Authors · 2025-12-03
>
> Thank you for your constructive feedback. We are glad to hear your appreciation of our ideas of hybrid combination of ML for SAT and asynchronous design. We hope to address your concerns and questions below.
>
> > [W1] engineering vs ML innovations.
>
> We would like to clarify that, while engineering efforts and system design is crucial to achieve practical wall-clock time reduction (demonstrated in Section 6.3 and 6.4), we have also proposed ML innovations that significantly improved the quality of neural heuristics and the success rate of standalone ML solving (demonstrated in Section 6.1 and 6.2). These innovations include integration of standalone ML solvers and ML-assisted solvers (Section 3.1), inclusion of learned clauses and partial assignment (Section 3.2), introduction of global attribute and recurrent updates (Section 3.3). As shown in Table 1 and Table 2, for GGNN architecture in average, these ML innovations improved the reduction rate $r$ from $4.84\times$ to $5.29\times$ and the standalone success rate $p$ from $20.7\%$ to $70.9\%$.
>
> > [W2/W5]  improvement expected since more compute resources are used, similar to portfolio approaches.
>
> We would like to reiterate that, since the CDCL algorithm is inherently sequential, the challenge is less about getting more compute resources, but more about how to better exploit compute resources to accelerate it, especially using GPUs. While portfolio approaches leverage multiple solver instances on multiple CPUs, our approach is inherently different, as we assist one CDCL solver with asynchronous ML inference on GPUs, which provides a different solution to the challenge.
>
> > [W3] augmentation with learnt clauses has been studied in G4SATBench.
>
> We would like to clarify that MAS-SAT integrates learned clauses in a different setup as G4SATBench [1] that has not been studied before.  In G4SATBench, learned clauses are added to the graph representations after the solver finishes to augment training data. However, in MAS-SAT, learned clauses are observed and updated for every interaction, which reflects the dynamics of learning of solvers.
>
> > [W4] no comparison with other GNN-based SAT solving.
>
> We would like to reiterate that our proposed MAS-SAT framework is compatible with all GNN architectures and previous paradigms of using ML for SAT solving. For controlled evaluation, we have performed extensive experiments on 7 datasets and 3 GNN architectures with the goal of comparing with previous GNN-based SAT solving methods at the framework level. Experimental results have shown that MAS-SAT demonstrated improved performance regardless of the specific dataset and GNN architecture.
>
> > [Q] clarifications about metrics used.
>
> The metrics $n$ and $r$ are introduced in Line 352-354. We perform one message-passing between each interaction, so numerically $n$ equals the number of interactions between the CDCL solver and the ML agent. In a more general sense it measures the computational cost of ML inference. $r$ is defined in Eqn. 12 and it indicates that at least half of the instances achieve the reduction. These two metrics measure the quality of ML models and can be translated into improvement in running time, which we have also reported in Table 4.
>
> [1] G4SATBench: Benchmarking and Advancing SAT Solving with Graph Neural Networks. TMLR 2024.

---

### Meta-Review · Area_Chair_BFNe · 2026-01-03

**Summary:**

The primary concerns expressed by the reviewers are:

(1) the machine learning innovations of the work are limited and the reported improvements are driven more by engineering effort.

(2) The experimental comparison does not provided sufficient evidence of the claimed outperformance of the method. Reviewers expressed concerns about lack of experimental detail, unprovided runtimes, absence of detailed analysis of the results, and the limited number of benchmarks.

(3) more compute resources are provided in parallel in the “asynchronous” setting, so a performance improvement is expected, and the comparison to other methods that are not provided with parallel compute is unfair. The approach is potentially more related to portfolio approaches, since it involves two different types of approach running in parallel, and therefore there should be some experimental comparison to a selection of portfolio techniques.

(4) The paper does not provide evidence that the approach can scale.

**Reviewer Concerns:**

Concern (1) is more of a matter of opinion, and the authors provided a rebuttal by pointing to the sections that provided a machine learning innovation, without expanding at all on why these constituted a significant methodological innovation. The text devoted to novel machine learning approaches in these sections is very limited (the sections together amount to 1 page, and at most half of that is technical contribution). The response could have been much more thorough to provide an expanded and more detailed discussion of the innovations, with a clearer explanation of their significance and difference from prior work.

Concern (2). The revision provided a survival plot for one dataset and an additional table. There is no additional analysis or discussion. Reviewers requested more information such as scatter plots and per-problem details. The survival plot indicates that the performance improvement is small; given this, there is a need for additional supporting evidence, either through experimentation on other benchmarks or a careful statistical analysis.

Concern (3). The authors argue that the "the CDCL algorithm is inherently sequential, the challenge is less about getting more compute resources, but more about how to better exploit compute resources to accelerate it, especially using GPUs". This argument needs more careful support. The authors are essentially arguing that there are parallel compute resources that can be used to support one type of algorithm but not another (i.e., you can run the GNN but not another parallel CDCL thread or another solver). The paper would need to include a careful explanation of the compute model and a justification of the claimed restrictions.

Concern (4). The response claims compatibility but does not provide evidence of successful integration.

**Reviewer Scores:**

Reviewer k2hd: VERY UNLIKELY TO CHANGE. The authors did not make any changes to the paper to address Reviewer k2hd's points and instead chose to provide rebuttals. The key criticism of the review was that the paper did not provide substantial machine learning innovations but instead represented primarily engineering improvements. The authors did not expand in their response on the machine learning innovations but instead specified the sections. The identified methodological sections are very brief. It is unlikely that the brief response would change the reviewer's opinion of the score.

Reviewer gowR. UNLIKELY TO CHANGE. The reviewer requested the reporting of absolute runtime and expressed a concern that the very small outperformance on one benchmark could be noise. In response, the authors included a survival plot. There is no statistical analysis of the results. The survival plot still displays the data in highly aggregated form and it remains difficult to determine if there is a meaningful or practically significant performance difference. There are stopping times in the survival plot where no outperformance would be observed. The reviewer requested a cactus plot or scatter plot, which would be more informative; based on the survival plot, there is a suspicion that such a plot would convey minimal advantage. Overall, the response is unlikely to change the reviewer’s mind that the performance evaluation does not provide adequate evidence of a genuine (and important) performance improvement.

Reviewer Pip4. VERY UNLIKELY TO CHANGE. The reviewer already gave a score of 6 but expressed several concerns that are unaddressed. These include the scalability of the approach (the authors respond by claiming that the approach is compatible but do not provide any experimental results) and the relatively small performance improvement. There is a request for a per-problem runtime performance comparison; this is not provided in the revised paper. There is a concern that the comparison should potentially be to portfolio methods, since the proposed technique uses additional parallel compute. The authors provide additional results in their response for other hybrid approaches and show outperformance, but the details of this experiment are not provided, and it remains unclear whether other algorithms are given the same opportunity to use additional parallel compute.

Reviewer 727G. UNLIKELY TO CHANGE. The reviewer’s main criticism is “the lack of detailed, reproducible experiments on standard benchmarks”. The response does not provide results for any other benchmarks. For the one competitive benchmark that is provided, the only additional results provided are a single survival plot and a summary table. The revisions are unlikely to change the reviewer’s opinion that there is inadequate experimental support for the claims of the paper.

---

### Decision · Program_Chairs · 2026-01-26

Reject